# Analysis of Factors Contributing to State Body Appreciation during Exercise

**DOI:** 10.3390/bs14080690

**Published:** 2024-08-08

**Authors:** Migle Baceviciene, Kristina Bliujute, Rasa Jankauskiene

**Affiliations:** 1Department of Physical and Social Education, Lithuanian Sports University, 44221 Kaunas, Lithuania; 2Department of Coaching Science, Lithuanian Sports University, 44221 Kaunas, Lithuania; k.bliujute@gmail.com; 3Institute of Sport Science and Innovations, Lithuanian Sports University, 44221 Kaunas, Lithuania; rasa.jankauskiene@lsu.lt

**Keywords:** state mindfulness, state body appreciation, state body surveillance, intrinsic motivation, exercise, obesity

## Abstract

This cross-sectional study aimed to analyse the factors contributing to state body appreciation (SBA) during exercise. After providing their informed consent, 200 study participants (mean age 30.0 ± 9.4 years, 72.0% were men) filled in an online questionnaire immediately after the completion of resistance training (n = 125), cardiovascular exercise (n = 55), or functional/interval group exercise (n = 20) sessions. The study measures included socio-demographic variables, body mass index (BMI), the duration of involvement in sports, SBA, state body surveillance (SBS), state mindfulness in physical activity, state intrinsic exercise regulation, and perceived pleasantness during exercise. The results showed that exercisers involved in sports for >2 years and those whose body weight was within a healthy range (<25.0 kg/m^2^) demonstrated higher SBA and mindfulness during exercise, lower SBS, more intrinsic exercise regulation, and higher satisfaction during physical activity compared to exercisers with less exercise experience (≤2 years) and a body weight higher than a healthy range (≥25.0 kg/m^2^). The negative effects of being overweight or obese were more pronounced in individuals with ≤2 years of sports participation, except regarding body surveillance and monitoring the mind. The factors significantly contributing to SBA during the exercise sessions were a longer duration of involvement in sports, a lower BMI and SBS, and greater mindful body acceptance and exercise pleasantness. Decreasing SBS and enhancing mindful body acceptance, pleasantness, and intrinsic motivation during exercise might significantly contribute to SBA in physical activity. These results can inform physical-activity-based programmes aiming to promote a positive body image. Also, these results show that it is important to increase education and develop competencies for fitness coaches to create inclusive and positive-body-image-promoting sports environments.

## 1. Introduction

### 1.1. The Importance of Positive Body Image for Exercise Adherence

Physical activity has many physical, psychological, and social benefits [1]. Nevertheless, more than one in three adults in the European Union are not sufficiently physically active, and sedentary behaviours are prevalent [2,3,4]. Thus, the factors that influence physical activity need to be identified. One of the factors that might increase physical activity is a positive body image [2,5,6,7]. Body image is a mental representation of the body’s appearance and functionality. It might be evaluated in the perceptual, cognitive, affective, and behavioural domains [7]. The associations between body image and physical activity are bidirectional. Physical activity participation is associated with and positively affects body image [8]. However, some specific contexts (e.g., mirrored walls of wellness centres or the appearance-focused language of fitness coaches) of physical activity might be variables that increase body anxiety and/or body dissatisfaction [9,10]. Further, a negative body image (body dissatisfaction or body shame) might impact physical activity from short- and long-term perspectives [5,7,11]. However, according to research, a positive body image is associated with higher physical activity and is greater in athletes and exercisers than in sedentary populations [5,12,13]. 

The central construct of positive body image is body appreciation. Body appreciation is defined as accepting one’s body regardless of its appearance and size or deviations from sociocultural appearance-related stereotypes, and respecting and loving one’s body; it is also the ability to reject sociocultural pressures regarding appearance and engage in health-promoting behaviours, including intrinsically motivated physical activity and exercise [6]. Body appreciation is associated with greater psychological functioning, higher self-esteem, self-compassion, mindfulness, and more intrinsic exercise motivation [14,15,16]. To date, body appreciation in sports contexts has been assessed as a stable, trait-like construct. However, scholars have drawn attention to the fact that body appreciation might fluctuate across time and situational contexts [17]. One of these contexts is sport and exercise. Bearing in mind the complex and bidirectional associations between physical activity and body image, it is important to understand what factors affect state body appreciation during physical exercise. To the best of our knowledge, there are very few studies aimed at assessing state body appreciation (SBA) during exercise, and the factors that contribute to SBA during physical activity are not well examined. Understanding these factors might inform exercise-based interventions that aim to promote a positive body image and adherence to physical activity.

### 1.2. Factors Contributing to Body Appreciation during Exercise

Based on the objectification (OT), self-determination (SDT), and monitor and acceptance theories (MAT) [18,19,20], among the most influential physical-activity-related factors that might affect SBA during exercise are self-objectification (body surveillance), the quality of exercise motivation, and mindfulness during exercise. Considering these theories, we will further discuss these factors in more detail. 

According to the OT [18], body self-objectification occurs when women internalise the sexual objectification of their bodies, leading them to view themselves as an object represented by how their bodies appear to others. Self-objectification mainly affects women; however, men might also be vulnerable [21]. Self-objectifying individuals engage in habitual, self-conscious body surveillance, which is described as the mental scanning of the body from an observer perspective [22]. In sports and exercise environments, especially those with mirrored walls, when wearing body-revealing sports apparel, and in environments in which fitness coaches use appearance-directed language, self-objectification and body surveillance might be increased [9,10,23]. Body surveillance draws exercisers’ attention towards external experiences (the constant checking of one’s appearance in the mirror or thinking about one’s appearance from a third person’s perspective when exercising) rather than drawing their attention towards internal body experiences and physical activity itself. Among the outcomes of state objectification during exercise are increased anxiety, reduced flow experiences, and awareness of internal body states [24]. According to the integrative process model of state self-objectification [24], these outcomes might be triggered by appearance monitoring, the experience of deviations from appearance standards, activating sex object schema, and the experience of stereotype threats. Further experience of deviations from appearance standards might activate appearance-related motivations [24]. Body objectification is associated with lower levels of body functionality appreciation, a main facet of positive body image [25]. However, a limited number of studies have assessed how state body surveillance is related to SBA during exercise other than yoga-based physical activities, and the present study aims to provide more empirical data on this topic.

The results of previous studies showed that body surveillance during exercise decreases enjoyment and pleasure during physical activity [26]. Affective judgements (pleasure and enjoyment) during physical activity are some of the strongest predictors of physical activity [20,27]. Thus, it is important to assess body surveillance as a factor that might negatively impact positive body image during physical activity and future engagement in exercise. 

Based on SDT, previous studies showed that the associations between physical activity and positive body image also depend on the quality of motivation. Appearance goals for exercise are consistently associated with a negative body image [7,20]. Intrinsic exercise motivation is the moderator of the associations between body appreciation and physical activity. A study by Homan and Tylka (2014) showed that, while exercise frequency was related to higher trait body appreciation, high levels of appearance-based exercise motivation weakened this relationship [14]. Also, controlled forms of exercise regulation (especially guilt-related exercise motivation) mediated the associations between appearance-based exercise goals and body appreciation. However, only a limited number of studies have assessed the associations between intrinsic exercise motivation and body appreciation during physical activity, and we aim to provide more knowledge on this topic. 

According to MAT, mindfulness (attention to the present moment, awareness of internal and external stimuli, and their non-judgemental acceptance) leads to greater psychological functioning [19]. Mindfulness during exercise (mindful monitoring and unconditional acceptance of the body and mind) is associated with a more positive body image [28]. There are types of physical activity that may be particularly effective in drawing exercisers’ focus to the internal experience of the body and mind. Examples of these activities are yoga, Tai Chi, qigong, Feldenkrais, and other modern forms of physical activity such as Pilates and Nirvana Fitness, which draw the attention of exercisers towards mindful breathing and physical sensations [29]. An increasing number of studies show that yoga positively affects body image through positive changes in body–self connection, body surveillance, and increased mindfulness during exercise [30]. However, the results of a previous cross-sectional study on a large sample of students participating in various sports suggested that mindfulness during physical activity (monitoring and acceptance of the body) mediated the associations between physical activity and body appreciation [31]. Thus, it is important to assess the role of mindfulness in forms of activity other than yoga-based ones, including various types of modern exercise such as resistance training, cardiovascular training workouts, and group fitness activities. Finally, it should also be noted that our understanding of the role of state mindfulness in SBA during physical activity as an immediate experience in exercise is also very limited [32]. A positive experience of pleasure, satisfaction, and mindfulness during physical activity and SBA might further develop into the trait of a positive body image and prolonged and mindful exercise participation. 

### 1.3. The Importance of Body Appreciation, Body Surveillance, Enjoyment, and Mindfulness during Physical Activity for Exercise Adherence in Higher Body Mass Exercisers

Increasing positive affective responses and decreasing body surveillance are extremely important for exercisers with a higher body weight. Previous studies have shown that people with a higher body weight express greater gym avoidance, higher body-related anxiety, self-monitoring of body weight, lower exercise self-efficacy, and face body-related stigma during exercise [33,34]. Social physique anxiety moderates the associations between body weight and exercise avoidance. Specifically, a previous study showed that body weight was associated with a desire to avoid exercise among individuals with high social physique anxiety, but not those with low social physique anxiety [35]. Thus, it is important to decrease appearance-related stress in people with a higher body mass to achieve a greater adherence to physical activity. A previous experimental study reported that women with a higher body mass showed greater perceived exertion than women whose body weight was within the healthy range during two 20 min sessions of treadmill exercise (one at a self-selected speed and one at an imposed speed). Neither group differed in their ratings of pleasure–displeasure during the session at the self-selected speed, but only the overweight women showed a significant decline in pleasure when the speed was imposed [36]. These results suggest that the skills of self-regulating the intensity of physical activity and noticing and accepting interoceptive and exteroceptive signals from the body might be increased if individuals possess mindful attention to and acceptance of the internal signals of the body. Thus, mindfulness during physical activity, with an attentional focus on internal body signals rather than appearance, might help people with a higher body mass to increase their skills of self-regulating exercise intensity and enjoying physical activity. However, body surveillance and mindfulness as factors contributing to an immediate experience of a positive body image and enjoyment during physical activity have never been tested before in exercisers with different body masses. 

### 1.4. The Present Study

The present study aimed to test the factors contributing to SBA during exercise in a sample of adult exercisers. Additional objectives were to compare the study variables between groups of different durations of involvement in sports and body mass indexes (BMIs), and to test the interaction between the duration of involvement in sports and BMI while controlling for sex and age. We expected negative associations between state body surveillance and SBA, and positive relationships between SBA and the other tested variables. Also, we hypothesised that all study variables (except state body surveillance) would be greater in exercisers with longer exercise experience and in exercisers whose body mass was within the healthy range. 

## 2. Materials and Methods

### 2.1. Study Design and Participants

The study was organised as cross-sectional. The study participants (n = 200) were exercisers at two wellness and sports centres who completed the survey up to 10 min after their exercise session. The inclusion criteria were as follows: Lithuanian-speaking adults aged ≥ 18 years who completed their exercise session no longer than 10 min before. The age of the study participants ranged from 18 to 57 years (mean 30.0, SD = 9.4 years); the majority of the study sample (72%) were men. The mean BMI of the exercisers was 25.5 kg/m^2^ (SD = 4.1), with a range of 17.6–41.5 kg/m^2^. Detailed sample characteristics are provided in Table 1.

### 2.2. Setting and Procedures

This study was approved by the Lithuanian Sports University Social Research Ethics Committee (Protocol No. SMTEK-131, 30 October 2022) and implemented in June–November 2023 via the free online platform Google Forms. The study was funded by the Research Council of Lithuania (Grant No. S-MIP-22-25).

The study was implemented in two wellness and sports centres, asking participants to complete the survey after an exercise session. The types of exercise sessions attended were resistance training; cardiovascular exercise on a treadmill or bicycle; and group fitness exercise (functional or interval training). Resistance training (also called ‘strength training’ or ‘weight training’) is the use of resistance to muscular contraction to build strength, anaerobic endurance, and the size of skeletal muscles. Functional training is a strength-based exercise that emphasises multi-joint movements that mimic everyday movement patterns. Functional training aims to enhance muscle strength and fitness for exertion during daily routine activities, and some of its most popular exercises include squats, lunges, push-ups, and pull-ups. Interval training is a type of training exercise that involves a series of high-intensity workouts interspersed with rest or break periods. The high-intensity periods are typically at or close to anaerobic exercise, while the recovery periods involve activity of a lower intensity. Interval training consists of a series of repeated rounds of exercise, ranging from several minutes to just a few seconds. During each interval, exercisers work at a set intensity for a specific period of time or distance (work interval) and follow this with a low-intensity recovery period (recovery interval). The duration of one exercise session in which the exercisers participated ranged from 45 min to 1.5 h.

Before the completion of the study survey, the participants were familiarised with the study aims, the content of the survey, the approximate duration of the survey, and information on data anonymity. The survey was anonymous, with no collection of personal data that could serve to identify the study participants. Next, the respondents were instructed by one of the authors (K.B.) on how to fill in the form correctly, and they provided digital consent to participate by clicking the option ‘I agree to participate’. Also, at any point in the survey, the respondents could stop by closing the browser without recording their answers. The participants filled out these electronic questionnaires in the lobby of the sports centres. It took 20–30 min to fill out the questionnaires.

### 2.3. Study Measures

Sociodemographic variables contained information on sex, age, involvement in sports and exercise duration in years, and the type of exercise session attended. Also, the responders provided information on their weight (kg) and height (cm), and their BMI was calculated using the equation: weight, kg/height, m^2^. Further, for descriptive purposes, the study sample was categorised into normal BMI (<25.00 kg/m^2^), overweight (25.00–29.00 kg/m^2^), and obese (≥30 kg/m^2^) groups.

The State Body Appreciation Scale (SBAS) is a psychometric tool designed to measure an individual’s momentary or state-level appreciation of their body [17]. The SBAS uses a Likert scale format, where respondents rate their agreement with each statement on a scale ranging from 1 ‘Strongly Disagree’ to 7 ‘Strongly Agree’. The Lithuanian translation of the tool was used in our previous study and the unidimensional factor structure was confirmed [37]. The psychometric characteristics of the trait Body Appreciation Scale 2 (BAS-2) in the Lithuanian population were good [12]. In this study, the indices of the internal consistency of the scale were good: Cronbach’s α = 0.981 (95% CI = 0.977–0.985), McDonald’s ω = 0.982 (95% CI = 0.979–0.986).

The State Body Surveillance (SBS) subscale from the Objectified Body Consciousness Scale (OBCS) was used to test body surveillance [38]. To focus on state body surveillance experience during physical activity rather than trait body surveillance, the Body Surveillance subscale of the OBCS was adopted. Seven items were employed for this purpose, aligning with a previous study that used the instrument to measure state body surveillance during physical activity [28]. Responses were recorded on a 7-point Likert-type scale, ranging from 1 (strongly disagree) to 7 (strongly agree), with higher values indicating greater state body surveillance. The Lithuanian version of the Body Surveillance subscale demonstrated good psychometric properties [39]. In this study, the internal consistency for the SBS was also excellent: Cronbach’s α = 0.953 (95% CI = 0.942–0.962), McDonald’s ω = 0.954 (95% CI = 0.944–0.964).

The State Mindfulness Scale in Physical Activity 2 (SMS-PA-2) is a psychometric instrument designed to measure state mindfulness during physical activity. Developed by Ulrich and colleagues, this scale consists of 19 statements attributed to four subscales: Monitoring Mind (MM, n = 6), Monitoring Body (MB, n = 6), Accepting Mind (AM, n = 3), and Accepting Body (AB, n = 4) (28). Each item on the SMS-PA 2 is rated on a Likert scale ranging from 1 (not at all) to 5 (very much), allowing participants to indicate the degree to which each statement reflects their experience during physical activity. Each subscale score is calculated by averaging the response options, with a higher score indicating higher mindful awareness and acceptance of the body and mind during physical activity. The psychometric properties of the Lithuanian translation of the SMS-PA-2 were confirmed in our previous study on physically active young adults [40]. The internal consistency of the SMS-PA-2 subscales in this study was also good. Cronbach’s α and McDonald’s ω coefficients for the SMS-PA-2 subscales are listed in the same order provided above: MM Cronbach’s α = 0.927 (95% CI = 0.910–0.942); McDonald’s ω = 0.929 (95% CI = 0.914–0.945); MB Cronbach’s α = 0.915 (95% CI = 0.895–0.932); McDonald’s ω = 0.917 (95% CI = 0.899–0.935); AM Cronbach’s α = 0.806 (95% CI = 0.745–0.854); McDonald’s ω = 0.806 (95% CI = 0.705–0.876); AB Cronbach’s α = 0.939 (95% CI = 0.923–0.951); McDonald’s ω = 0.939 (95% CI = 0.926–0.953).

State Intrinsic Regulation was tested by the Behavioural Regulation in Exercise Questionnaire 2 (BREQ-2) [41] subscale, adopting it to reflect the state experience of pleasure during the exercise. The State Intrinsic Regulation subscale consists of four statements rated on a 5-point Likert scale from 1 ‘not true’ to 5 ‘very true’. The averaged response options reflect a total score, which indicated more state intrinsic regulation of exercise during the last session. The factor structure of the BREQ-2 was confirmed in our previous study [42]. In this study, the internal consistency of the Intrinsic Regulation subscale was satisfactory: Cronbach’s α = 0.962 (95% CI = 0.954–0.971); McDonald’s ω = 0.962 (95% CI = 0.952–0.970).

The Empirical Valence Scale (EVS) [43] was used to assess pleasantness during exercise. A single question was asked to rate the pleasantness of the previous exercise session on a scale from −10 (very unpleasant) to 10 (very pleasant), with 0 as a neutral point.

### 2.4. Sample Size

The convenience sampling method was used and based on voluntary participation. First, to define the adequacy of the sample size, it was established that, to obtain statistical significance between two means with 80% power, a medium effect size of the value 0.5, α < 0.05, and the application of a two-tailed test, approximately 65 study subjects per group were required. For the linear regression, the required sample size for a given effect size of 0.5, power of 80–99%, and level of significance α = 0.01, 51–104 subjects were needed [44]. 

### 2.5. Statistical Analysis

Descriptive statistics were computed to summarise the characteristics of the study sample. For continuous variables, means and standard deviations (SDs) were calculated. Skewness and Kurtosis values were examined to assess the normality of the distribution for each continuous variable. For the categorical variables, frequencies and percentages were calculated. To assess the internal consistency of the study measures, Cronbach’s α and MacDonald’s ω were calculated [45,46].

Further, Multivariate Analysis of Covariance (MANCOVA) was employed with state body appreciation, exercise motivation, exercise pleasantness, and mindfulness as dependent variables, BMI and the duration of involvement in sports groups as fixed factors, including interaction in between, and age and sex as covariates. Partial eta-squared represented the effect size and was classified as small (above 0.01 and below 0.06), medium (above 0.06 and below 0.14), and large (≥0.14) [47]. Finally, a hierarchical linear regression analysis was performed to predict SBA, with the variables entered in blocks to assess the incremental variance explained by each set of predictors. Variance inflation factors (VIFs) were calculated to assess multicollinearity between independent variables. VIF = 1 indicates no correlation between the predictor and other variables, and no multicollinearity, 1 < VIF < 5 indicates a moderate correlation, but is generally acceptable, and VIF ≥ 5 suggests a high correlation and potential multicollinearity [48]. 

All statistical analyses were conducted using the free software JASP v.0.18.3 obtained from the official website (JASP Team: Amsterdam, The Netherlands). A *p*-value of less than 0.05 was considered to be statistically significant.

## 3. Results

The sample characteristics are presented in Table 1. The mean age of the study participants was 30.0 ± 9.4 years. Most of the study participants were men (72%) with a normal body mass index (51%) who attended the muscle power exercise session (62.5%). Also, for further comparisons, the responders were divided into two groups according to their duration of involvement in sports: ≤2 years (46.0%) and >2 years (50%), while 4% could not recall this duration and did not specify it.

Table 2 presents a multivariate analysis examining the effects of BMI and the duration of involvement in sports on state body appreciation, state mindfulness during exercise, pleasantness during exercise, and state exercise motivation, with age and sex included as covariates. The analysis indicates that being overweight or obese negatively affected state body appreciation, body surveillance, mindfulness, pleasantness during exercise, and intrinsic exercise motivation, with no significant impact on monitoring the mind during exercise. Conversely, participating in sports for more than two years positively and significantly affected all measured outcomes. Most of the effects demonstrated medium effect sizes. Additionally, a significant interaction between BMI and the duration of sports involvement suggests that the negative effects of being overweight or obese were more pronounced in individuals with two or fewer years of sports participation, except in the domains of body surveillance and monitoring the mind. Visual representations of the significant effects of BMI, the duration of sports involvement, and their interaction on state body appreciation, mindfulness, and exercise motivation are provided in Appendix A, Figure A1, Figure A2, Figure A3, Figure A4, Figure A5 and Figure A6.

Hierarchical linear regression was conducted to determine the factors significantly contributing to SBA during the exercise sessions (Table 3). VIFs ranging from 1.08 to 2.97 did not detect a multicollinearity problem and indicated the models’ stability and reliable estimates. Step 1 presents the effects of sociodemographic factors, explaining 14% of the variance of the SBA, with a positive and significant effect of a longer duration of involvement in sports. Step 2 demonstrates the effects of body-image-related factors on SBA, increasing the explained variance up to 58%: a higher BMI and state body surveillance had negative effects on the SBA. In Step 3, SMS-PA-2 subscales were added as independent variables, with the only significant effect of mindful acceptance of the body during the exercise increasing R^2^ to 0.66. The final Step 4 additionally revealed the significant and positive effect of pleasantness during exercise. The final model explained 71% of the variance of the SBA. The effects of the duration of the involvement in sports, BMI, state body surveillance, and state mindful body acceptance remained significant throughout all the steps, indicating their independent effects. A significant change in R^2^ at each step provides evidence that the new set of predictors meaningfully contributes to explaining the variance in the SBA.

## 4. Discussion

### 4.1. Factors Contributing to Body Appreciation during Exercise

Based on OT, SDT, and MAT, the present study aimed to analyse the factors contributing to SBA during exercise in a sample of exercisers practising resistance, cardiovascular training, and functional and interval group training activities. In line with our hypothesis, the analysis of hierarchical linear regressions revealed that body surveillance and body mass index were negatively associated with body appreciation, and this model explained 58% of the variance. The analysis of further steps showed that body surveillance remained the main factor negatively contributing to SBA. These results extend previous findings based on OT [18] that have consistently reported that body surveillance is negatively related to trait body appreciation [16]. These findings are in line with previous studies that have shown that body surveillance during exercise is negatively related to physical self-concept and the psychological benefits of exercise [26,49,50]. Our study provides important new evidence that constant control of one’s appearance and focusing attention on one’s appearance are associated with lower body comfort and good feelings about the body during exercise. This finding suggests that the focus of attention of exercisers during exercise is an important variable that should be controlled in physical activity programmes.

Further, our research showed that pleasantness, mindful acceptance of the body, and intrinsic motivation during exercise were the main positive contributors to body appreciation during physical activity. Importantly, the final model explained 71% of the variance in SBA. Specifically, this means that exercising for pleasure and enjoyment, satisfaction and pleasantness during physical activity, and mindful acceptance of the body during exercise are positively related to respecting one’s body, feeling attractive, and showing attention to the body’s needs when exercising. These findings are in accordance with the literature on positive body image and the developmental theory of embodiment, suggesting that positive embodiment is promoted when people participate in enjoyable and mindful exercise activities, express attentiveness to the body’s internal and external signals, and experience a sense of flow and use exercise for self-care, but not for body or appearance control [51,52,53,54]. The present study provides important new empirical knowledge that these associations exist not only between general (trait), but also between state variables. Prolonged positive cognitive and affective experiences during physical activity might be a background for the development of a general positive body image and engagement in health-related behaviours, including mindful physical activity [55]. Future studies should test whether state body image, mindfulness, and exercise motivation-related variables in exercise might predict trait body appreciation and exercise adherence. 

### 4.2. Analysis of Factors Contributing to Body Appreciation during Physical Activity in Exercisers of Different Exercise Experiences and Body Masses 

In the present study, we found that body surveillance during physical activity was higher in exercisers with lower exercise experience. This finding is difficult to explain, since the present study is cross-sectional. Previous findings have shown that, in general, physical activity might positively affect body image through an increase in body functionality appreciation, which is the opposite to self-objectification [22,25]. According to previous studies, state body surveillance might negatively affect pleasantness and enjoyment in exercise and intentions to exercise [50,56], therefore, exercisers who do not feel pleasure during physical activity might be those who quit exercising more frequently [57]. Future studies using designs that are not cross-sectional should test our findings. 

We also observed that body surveillance during exercise was significantly higher in individuals with a higher body mass. This finding is in accordance with previous findings suggesting that individuals with a higher body weight report a lower body appreciation compared to those individuals whose BMI is within the healthy range [58]. However, our study provides important evidence that, compared to exercisers with a healthy body weight, higher-body-mass exercisers more frequently focus their attention on their appearance during physical activity and perceive a lower acceptance of their bodies during exercise. 

In the present study, we found that body appreciation, mindfulness during physical activity, pleasure during exercise, and intrinsic motivation were significantly greater in experienced exercisers (>2 years) compared to exercisers with less experience, and in exercisers whose body weight was within the healthy range compared to those with a higher body weight. Previous studies have also suggested that pleasure and enjoyment during exercise are related to greater physical activity engagement [57]. Also, experienced exercisers report more mindfulness during exercise compared to less experienced exercisers [59]. The findings of previous studies have also suggested that exercising for pleasure and enjoyment is associated with prolonged exercising [20]. The present study provides the important empirical knowledge that internal motivation, pleasure, and mindfulness (monitoring and accepting the body and mind) during exercise and body appreciation during exercise are variables that differentiate people with longer exercise experience from novice exercisers and people who are overweight from those whose body weight is within the healthy range. Previous studies have also shown that overweight individuals experience higher body-related anxiety and self-monitoring of their body weight during exercise [33,34,35]. In future studies, it is important to further investigate these factors in studies with experimental and longitudinal designs to understand the mechanisms through which positive changes in SBA occur. 

Finally, we also found that the negative effect of increased body mass on body appreciation, mindfulness (excluding mind monitoring), intrinsic motivation, and pleasantness during exercise was greater for exercisers with less exercise experience. In other words, the negative effects of being overweight or obese on body appreciation, mindfulness, motivation, and pleasantness during exercise were more pronounced in people with fewer years of exercise participation (two years and less in our study). It seems that individuals with greater exercise experience and a high body mass appreciate their bodies more, are more mindful and internally motivated during physical activity, and feel more pleasure compared to less experienced individuals with a higher body mass. Importantly, no interaction between BMI and exercise experience was observed for body surveillance, suggesting that exercise experience may not protect exercisers with a higher body mass from body surveillance in the same way as exercisers with a healthy body mass. 

### 4.3. Practical Implications

The present study has important practical implications. First, the results of the study can inform exercise-related positive body image promotion and obesity prevention interventions. In these interventions, it is important to control the focus of the attention of participants using education methods that help to decrease body surveillance during exercise. A previous study concluded that increasing positive affective judgements during physical activity might promote physical activity; however, these efforts might be limited if body surveillance is present [24]. It is also recommended to strengthen mindfulness during physical activity, since mindfulness might help to decrease body surveillance, increase pleasantness, and promote internal motivation during physical activity [49,55]. There is evidence that mindfulness during physical activity might also promote trait body appreciation [60]. Since individuals with a higher body weight report less favourable relationships with their bodies, lower mindfulness, less intrinsic motivation, and less pleasantness during physical activity, compared to exercisers with a healthy body mass, it is important to address these differences when prescribing and implementing exercise programmes. For overweight individuals, mindfulness and pleasantness during physical activity should be promoted, aiming to increase their intrinsic motivation, decrease body surveillance, and enhance a positive body image. It is also important to strengthen the education of sports and fitness coaches and other personnel seeking to develop their competencies to implement strategies and practical applications that help to increase mindfulness, internal motivation, and the focus of attention on internal body signals rather than appearance in exercisers. Sport-science-related studies should include teaching about body image and the development of safe and inclusive physical activity environments. 

### 4.4. Strengths and Limitations of the Study

One of the strengths of the present study is that we employed a sample of adults exercising rarely tested exercise types in mindfulness-focused research. The present study is one of the first studies to assess SBA and the contributing variables in resistance training, cardiovascular training, and group fitness activity contexts. Previous studies have shown that sports environments such as gyms might increase self-objectification in exercisers [10]. However, the factors that positively contribute to SBA had never been tested before in similar samples. Assessing the factors that positively contribute to body appreciation is one of the biggest priorities of body image and physical activity research [11,61]. Further, the findings of our study showed that mindfulness during physical activity had an important contribution to feelings of a positive body image during exercise. Previous experimental studies showed that yoga might positively affect trait body appreciation [30,60,62]. Our study adds the important knowledge that exercising through other types of physical activity with a higher physical activity intensity, such as resistance training, cardiovascular training, and functional group exercise, might be associated with a positive body image in the presence of mindfulness. Finally, an important strength of the present study is that our sample consisted of adult exercisers, not university students who typically participate in body-image-related research studies. 

Among its strengths, the present study has important limitations that should be discussed. First, the cross-sectional design prevents us from drawing conclusions regarding the impacts of the tested variables. Second, the sample is small and does not represent all exercisers, therefore, the generalisability of the findings is limited. The small sample size and limited statistical power prevented us from conducting deeper analyses. Future studies are recommended to test our findings in larger samples. Finally, the distribution of sexes was unequal, with men representing 70% of the sample. We could not compare the study variables between different exercise types involving the participants, since the number of participants was extremely unequal. Also, we calculated BMIs from self-reported body weight and height data. BMI is not an accurate measure for individuals involved in resistance training, since increases in BMI might be related to muscle mass hypertrophy. In future studies, we recommend measuring body mass objectively. Another limitation of the present study is that we did not assess the history of clinical body image concerns or eating disorders, which might influence the results. Finally, we did not assess previous experiences of mindfulness-based activities, which may also have an effect on the results. Future studies should address these issues.

## 5. Conclusions

Our findings represent an important extension given the limitations of research on SBA and its correlates in sports and physical activity. In the present study, we found that body surveillance was negatively associated and pleasantness, intrinsic motivation, and mindfulness (body acceptance) were positively associated with body appreciation during exercise in a sample of adult men and women involved in resistance, cardiovascular training, and group fitness exercise. Lower body surveillance and higher state body appreciation, mindfulness during physical activity, intrinsic motivation, and pleasantness during exercise were observed in individuals with longer exercise experience and a healthy body weight compared to those with less exercise experience and a higher body weight. The negative effects of being overweight or obese on state body appreciation, mindfulness, intrinsic motivation, and pleasantness during exercise were more pronounced in individuals with two or fewer years of sports participation (except for body surveillance and monitoring the mind). Decreasing body surveillance (changing the focus of attention from appearance control towards internal or external body signals and/or physical activity itself) and enhancing mindful body acceptance, pleasantness, and intrinsic motivation during exercise might significantly contribute to the development of state body appreciation during physical activity and exercise adherence. Also, these results show that it is important to increase education and develop competencies for fitness coaches to create inclusive and positive body-image-promoting sports environments.

## Figures and Tables

**Table 1 behavsci-14-00690-t001:** Sample characteristics (n = 200).

Characteristics		n (%)
Sex	Men	144 (72.0)
	Women	56 (28.0)
Age, years	m (SD), range	30.0 (9.4), 18–57
Body mass index, categorical	Normal	102 (51.0)
	Overweight	68 (34.0)
	Obese	30 (15.0)
Body mass index, kg/m^2^	m (SD), range	25.5 (4.1), 17.6–41.5
Involvement in sports, categorical	≤2 years	92 (46.0)
	>2 years	100 (50.0)
	Not specified	8 (4.0)
Involvement in sports, years	m (SD), range	4.8 (6.1), 0.5–41.0
Type of exercise attended	Resistance exercise	125 (62.5)
	Cardiovascular training	55 (27.5)
	Functional and interval training	20 (10.0)

n = frequency, m = mean, and SD = standard deviation.

**Table 2 behavsci-14-00690-t002:** The multivariate analysis of covariance (MANCOVA) of the effects of body mass index and the duration of involvement in sports on state body appreciation, body surveillance, state mindfulness, state exercise regulation, and pleasantness during the exercise, m (95% CI).

Study Measures	Involvement in Sports	BMI, kg/m^2^	Mean	95% CI	Effect of BMI, *ŋ*^2^*, p*	Effect of Involvement, *ŋ*^2^*, p*	Interaction BMI × Involvement, *ŋ*^2^*, p*
State Body Appreciation Scale	≤2 years	<25.0	3.70	3.45–3.95	0.12, <0.001	0.15, <0.001	0.03, 0.014
		≥25.0	2.69	2.41–2.96			
	>2 years	<25.0	4.14	3.88–4.40			
		≥25.0	3.78	3.53–4.04			
OBCS: State Body Surveillance	≤2 years	<25.0	3.08	2.67–3.49	0.06, <0.001	0.09, <0.001	0.01, 0.249
		≥25.0	4.09	3.64–4.54			
	>2 years	<25.0	2.38	1.96–2.81			
		≥25.0	2.90	2.49–3.31			
SMS-PA-2: Monitoring Mind	≤2 years	<25.0	3.33	3.08–3.59	0.01, 0.101	0.03, 0.012	0.00, 0.985
		≥25.0	3.11	2.83–3.39			
	>2 years	<25.0	3.67	3.41–3.94			
		≥25.0	3.45	3.20–3.71			
SMS-PA-2: Monitoring Body	≤2 years	<25.0	3.95	3.72–4.18	0.08, <0.001	0.07, <0.001	0.04, 0.006
		≥25.0	3.14	2.89–3.39			
	>2 years	<25.0	4.08	3.84–4.32			
		≥25.0	3.94	3.71–4.17			
SMS-PA-2: Accepting Mind	≤2 years	<25.0	3.08	2.81–3.35	0.03, 0.017	0.05, 0.004	0.03, 0.028
		≥25.0	2.42	2.12–2.72			
	>2 years	<25.0	3.19	2.91–3.47			
		≥25.0	3.16	2.89–3.43			
SMS-PA-2: Accepting Body	≤2 years	<25.0	3.65	3.36–3.94	0.09, <0.001	0.07, <0.001	0.03, 0.014
		≥25.0	2.64	2.32–2.95			
	>2 years	<25.0	3.84	3.54–4.17			
		≥25.0	3.58	3.29–3.87			
State intrinsic exercise regulation	≤2 years	<25.0	6.40	5.22–7.58	0.10, <0.001	0.10, <0.001	0.02, 0.035
		≥25.0	2.27	0.97–3.57			
	>2 years	<25.0	7.90	6.67–9.13			
		≥25.0	6.42	5.23–7.60			
Pleasantness during exercise	≤2 years	<25.0	4.11	3.84–4.37	0.08, <0.001	0.10, <0.001	0.04, 0.007
		≥25.0	3.19	2.90–3.48			
	>2 years	<25.0	4.37	4.10–4.64			
		≥25.0	4.20	3.94–4.46			

CI = confidence interval; *ŋ*^2^ = partial eta-squared, BMI = body mass index, OBCS = Objectified Body Consciousness Scale; SMS-PA-2 = State Mindfulness Scale for Physical Activity-2; sex and age are added to the MANCOVA model as covariates.

**Table 3 behavsci-14-00690-t003:** Hierarchical linear regression predicting state body appreciation during the exercise session (n = 200).

Study Measures	*B*	*β*	*t*	*p*	VIF
Step 1					
Female sex	0.25	0.11	1.46	0.147	1.18
Age, years	0.01	0.09	1.15	0.253	1.32
Involvement in sports, years	0.05	0.31	4.25	<0.001	1.13
Model summary: *F* = 10.03, *p* < 0.001, *R*^2^ = 0.14					
Step 2					
Female sex	−0.09	−0.04	−0.74	0.460	1.24
Age, years	0.01	0.07	1.29	0.198	1.33
Involvement in sports, years	0.03	0.17	3.19	0.002	1.20
Body mass index, kg/m^2^	−0.06	−0.22	−4.33	<0.001	1.08
OBCS: State Body Surveillance	−0.41	−0.62	−11.98	<0.001	1.17
Model summary: *F* = 50.60, *p* < 0.001, *R*^2^ = 0.58, ∆*R*^2^ = 0.44, ∆*p* < 0.001					
Step 3					
Female sex	−0.11	−0.05	−0.91	0.366	1.36
Age, years	0.01	0.09	1.82	0.071	1.36
Involvement in sports, years	0.03	0.15	3.07	0.002	1.23
Body mass index, kg/m^2^	−0.04	−0.16	−3.52	<0.001	1.12
OBCS: State Body Surveillance	−0.27	−0.41	−7.24	<0.001	1.72
SMS-PA-2: Monitoring Mind	0.01	0.01	0.21	0.835	1.74
SMS-PA-2: Monitoring Body	0.15	0.12	1.94	0.054	2.13
SMS-PA-2: Accepting Mind	0.01	0.01	0.09	0.932	1.82
SMS-PA-2: Accepting Body	0.26	0.27	3.77	<0.001	2.78
Model summary: *F* = 39.74, *p* < 0.001, *R*^2^ = 0.66, ∆*R*^2^ = 0.08, ∆*p* < 0.001					
Step 4					
Female sex	−0.10	−0.04	−0.86	0.390	1.37
Age, years	0.01	0.08	1.71	0.090	1.36
Involvement in sports, years	0.02	0.13	2.78	0.006	1.24
Body mass index, kg/m^2^	−0.03	−0.12	−2.82	0.005	1.15
OBCS: State Body Surveillance	−0.23	−0.35	−6.52	<0.001	1.81
SMS-PA-2: Monitoring Mind	−0.01	−0.01	−0.12	0.901	1.77
SMS-PA-2: Monitoring Body	0.04	0.03	0.48	0.630	2.33
SMS-PA-2: Accepting Mind	0.01	0.01	0.21	0.838	1.83
SMS-PA-2: Accepting Body	0.18	0.19	2.75	0.007	2.97
State intrinsic exercise regulation	0.13	0.13	1.90	0.059	2.70
Pleasantness during exercise	0.05	0.20	3.05	0.003	2.62
Model summary: *F* = 39.48, *p* < 0.001, *R*^2^ = 0.71, ∆*R*^2^ = 0.05, ∆*p* < 0.001					

*B*—unstandardised; *β*—standardised regression coefficient; *t*—*t*-test; VIF—variance inflation factor; OBCS = Objectified Body Consciousness Scale; SMS-PA-2 = State Mindfulness Scale for Physical Activity-2.

## Data Availability

The data set generated and analysed in this study is available from the corresponding author.

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
