# Peer review of "Analysis of Factors Contributing to State Body Appreciation during Exercise"

_behavsci, 2024, doi:10.3390/bs14080690_

Round 1

Reviewer 1 Report

Comments and Suggestions for Authors

The study is interesting because of the use of psychological variables involved in exercise. However, the relevance of the study and the variables involved is unclear. However, there are some methodological limitations that should be considered

Introduction

The introduction is very long and should stick only to the focus of the study and the types of exercise used in the study itself, as well as the changes that occur during exercise.

I missed the gap in the literature to which this study specifically contributes.

The authors should consider the relevance of the study in the introduction, it is not very clear why it is important to understand the variables that influence the body's state of appreciation during exercise.

It is also unclear how the variables analyzed can contribute to the body's state of appreciation

Methods:

Add total sample calculation

The sample analyzed should be better evaluated. Do people have a history of image disorders? Do they have a history of mental illness? Do they practice other exercises that can affect body appreciation, such as mindfulness or yoga?

Add the experimental design of the study.

The intra-individual assessment time is long (from 45 minutes to 90 minutes), so this should be taken into account in the statistical analysis.

Results

Dispersion gradients should be added

Discussion

Previous studies should be discussed

Author Response

Dear Reviewer,

Thank you for your time reviewing our paper and for your comments. All changes made in the text are highlighted in blue font.

Comments and Suggestions for Authors

The study is interesting because of the use of psychological variables involved in exercise. However, the relevance of the study and the variables involved is unclear. However, there are some methodological limitations that should be considered

Introduction

The introduction is very long and should stick only to the focus of the study and the types of exercise used in the study itself, as well as the changes that occur during exercise.

Thank you for your comment. As the topic analysed (body appreciation, mindfulness during exercise) may be new to readers, we have decided to divide the text into subsections focusing on specific topics for clarity and ease of reading.  We did not aim to analyse body image in different exercise types and we had a small sample with very different distribution numbers, we did not discuss the effect of different exercise types on body image in the Introduction.

I missed the gap in the literature to which this study specifically contributes.

The gap in the literature is presented in lines 64-68, 114-116, 134-138 and 163-165.

The authors should consider the relevance of the study in the introduction, it is not very clear why it is important to understand the variables that influence the body's state of appreciation during exercise.

Thank you for this comment. The gap in the literature is now presented in the first subsection: „1.1. The importance of positive body image for exercise adherence“.

It is also unclear how the variables analyzed can contribute to the body's state of appreciation

The text was double-checked and revised to address this issue.

Methods: Add total sample calculation

Sample size calculation is now added to the Methods section 2.4.

The sample analyzed should be better evaluated. Do people have a history of image disorders? Do they have a history of mental illness? Do they practice other exercises that can affect body appreciation, such as mindfulness or yoga?

Thank you for your comment. We did not assess previous body image concerns or body image-related psychological disorders in the present sample. We included this as a limitation of the study.

Do they practice other exercises that can affect body appreciation, such as mindfulness or yoga?

We did not assess whether our sample practised yoga or mindfulness-based activities, as we were testing the immediate effect of exercise on body image and related variables. However, we acknowledge that this may be a limitation of the present study and have included a sentence in the limitations section.

Add the experimental design of the study.

The study is cross-sectional. We added this information to the Methods section 2.1.

The intra-individual assessment time is long (from 45 minutes to 90 minutes), so this should be taken into account in the statistical analysis.

The duration of exercise sessions was 45 minutes – 1,5 hours. The respondents filled questionnaires immediately after exercise sessions and it took approximately 20-30 minutes. This procedure is presented in Methods.

Results

Dispersion gradients should be added

According to the Reviewer‘s 2 comments, t-test tables were replaced with a more advanced statistical analysis.

Discussion

Previous studies should be discussed.

Thank you for this comment. The novelty of the study limits our possibility to compare the findings directly. However, we discuss our findings in light of the theoretical background and previous studies based on these theories.

Reviewer 2 Report

Comments and Suggestions for Authors

This study aimed to investigate factors contributing to State Body Appreciation during exercise.

Taking in account grouping variables of sex, BMI and exercise experience is main strength of the research, but dividing subjects in only 2 groups according them is main weakness.  I understand authors have small sample size to establish more control over grouping variables which is a shame because the idea is promising.

Never the less, the research was carried out correctly, the applied techniques are adequate, adequate conclusions were drawn, and the references are adequate. In a desperate way, we came to some new knowledge about a very important topic, both for scientists and even more so for practitioners. This research could be considered as an initial research on this topic and as a justification to repeat it on a much larger sample.

In addition, instead of the t test, I would suggest MANOVA or MANCOVA in processing the data.

Author Response

Dear Reviewer,

Thank you for your time reviewing our paper and for your valuable comments. All changes made in the text are highlighted in blue font.

Comments and Suggestions for Authors

This study aimed to investigate factors contributing to State Body Appreciation during exercise.

Taking in account grouping variables of sex, BMI and exercise experience is main strength of the research, but dividing subjects in only 2 groups according them is main weakness.  I understand authors have small sample size to establish more control over grouping variables which is a shame because the idea is promising.

Thank you for this important comment. The analysis was improved by replacing the comparisons across 2 groups with the MANCOVA as recommended with the control of sex and age.

Never the less, the research was carried out correctly, the applied techniques are adequate, adequate conclusions were drawn, and the references are adequate. In a desperate way, we came to some new knowledge about a very important topic, both for scientists and even more so for practitioners. This research could be considered as an initial research on this topic and as a justification to repeat it on a much larger sample.

 Thank you for this comment. We included it as the limitation of the study and recommendation for future studies.

In addition, instead of the t test, I would suggest MANOVA or MANCOVA in processing the data.

Thank you for this important comment. We replace t-test tables with MANCOVA statistics with state body appreciation, body surveillance, exercise motivation, exercise pleasantness and mindfulness as dependent variables, BMI and the duration of involvement in sports groups as fixed factors including interaction in between whereas age and sex as covariates.

Reviewer 3 Report

Comments and Suggestions for Authors

I congratulate the authors for the nice article. I have some minor suggestions that could enhance the overal quality of the article

1. update some references with newer ones;

2. the introduction is quite general and wide. It could be revised, shortened and contextualised;

3. the article mentiones many used instruments (scales) though it would be very important to speak about their validation for the given population;

4. discussion part is interesting and provides quite some informations on the matter, though (for the articles sake) I would suggest to re-write it and reshapen in the form that it does not repeat the same issues over and over;

5. perhaps authors could provide some reccomendations in the conclusion part.

Author Response

Dear Reviewer,

Thank you for your time reviewing our paper and for your valuable comments. All changes made in the text are highlighted in blue font.

Comments and Suggestions for Authors

I congratulate the authors for the nice article. I have some minor suggestions that could enhance the overal quality of the article

Thank you.

  1. update some references with newer ones;

We double-checked the references and acknowledged that several references are old. However, other references are related to the theories, the development (validation) of the study measures or statistical analysis.

  1. the introduction is quite general and wide. It could be revised, shortened and contextualised;

Thank you for your comments. As the topic analysed (body appreciation, mindfulness during exercise) may be new to readers, we have decided to divide the text into subsections focusing on specific topics for clarity and ease of reading.

  1. the article mentiones many used instruments (scales) though it would be very important to speak about their validation for the given population;

Thank you for this comment. The majority of measures were initially validated in a samples of young adults (university students). Most of them were involved in resistance exercise or other leisure time exercise activities.

  1. discussion part is interesting and provides quite some informations on the matter, though (for the articles sake) I would suggest to re-write it and reshapen in the form that it does not repeat the same issues over and over;

Thank you for this comment. We revised the discussion and organized it according to the study‘s aim. We also divided the text into subsections. We also double-checked the text and deleted repetitive sentences.

  1. perhaps authors could provide some reccomendations in the conclusion part.

Thank you, we included a recommendation.